# Influence of the COVID-19 Pandemic on the Lifestyles of Health Sciences University Students in Spain: A Longitudinal Study

**DOI:** 10.3390/nu13061958

**Published:** 2021-06-07

**Authors:** Idoia Imaz-Aramburu, Ana-Belén Fraile-Bermúdez, Batirtze San Martín-Gamboa, Silvia Cepeda-Miguel, Borja Doncel-García, Ainhoa Fernandez-Atutxa, Amaia Irazusta, Idoia Zarrazquin

**Affiliations:** 1Department of Nursing I, Faculty of Medicine and Nursing, University of the Basque Country, 48940 Leioa, Bizkaia, Spain; idoia.imaz@ehu.eus (I.I.-A.); anabelen.fraile@ehu.eus (A.-B.F.-B.); batirtze.sanmartin@ehu.eus (B.S.M.-G.); silvia.cepeda@ehu.eus (S.C.-M.); ainhoa.fernandez@ehu.eus (A.F.-A.); amaia.irazusta@ehu.eus (A.I.); 2OSI Bilbao-Basurto, Osakidetza Basque Health Service, 48013 Bilbao, Bizkaia, Spain; borja.doncelgarcia@osakidetza.eus; 3Department of Nursing II, Faculty of Medicine and Nursing, University of the Basque Country, 20014 Donostia/San Sebastián, Guipuzkoa, Spain

**Keywords:** COVID-19, Mediterranean diet, physical activity, health science university students

## Abstract

The COVID-19 pandemic has significantly impacted daily activities worldwide. University students may have experienced substantial changes in daily living as a result of restrictions on university attendance. The return to normalcy may take a long time, and understanding the influence that shifts in daily routines have had on the lifestyles of university students may inform approaches to support overall well-being. We analyzed changes in the lifestyles of students enrolled at a health sciences university during the COVID-19 pandemic. This longitudinal study took place at the Faculty of Medicine and Nursing in the University of the Basque Country in Spain, and the final sample consisted of 113 nursing students, 109 medical students, and 45 physiotherapy students. Our results demonstrate changes in lifestyles of university students during the pandemic. MedDiet adherence scores and the percentage of students with high adherence increased during the pandemic. This increase was due to the increased consumption of vegetables and nuts. In terms of physical activity, the practice of moderate and intense physical activity was maintained. These results provide important information for both public health authorities and educational institutions to guide strategies to maintain the well-being of students and enhance opportunities for young adults to lead a healthy lifestyle.

## 1. Introduction

The emergence and expansion of the SARS-CoV-2 virus in late 2019 led to a global pandemic causing severe disease and numerous deaths worldwide. Throughout the evolution of the pandemic, governments around the world responded by implementing different measures to contain the viral spread, including instituting home confinement and travel restrictions, as well as limiting capacity in indoor spaces. UNESCO data indicate that, in early April 2020, these restrictions led to closures of schools and higher education institutions, including universities, in 185 countries. These closures affected 1,542,412,000 learners, constituting 89.4% of total enrolled learners [1]. 

In Spain, where the present study took place, the government-mandated home confinement of the population from March to April 2020 [2] as well as the closure of universities and higher education institutions. The fact that the universities were closed greatly impacted higher education institutions and their attending students. To minimize the impact of this situation on the ~1.5 million attending students, the Spanish university system adapted to allow students to finish their academic courses online [3]. In line with such changes, in September 2020, the University of the Basque Country in Spain began a new academic course using blended learning for health sciences students, combining in-person and remote classes. This model was implemented because, on the one hand, large class sizes were difficult to hold in person due to social distancing requirements and, on the other hand, specific practices discussed in these courses cannot be taught remotely and require in-person interaction. 

Such changes to university routines could have an effect on students. University life usually coincides with the transition between adolescence and adulthood, a period that is essential in establishing a healthy lifestyle that, in many cases, will be maintained throughout life [4]. In university students enrolled in health sciences programs, maintenance of healthy lifestyles takes on a special relevance, as these students will become health professionals who promote healthy lifestyles in the population. It is therefore important to analyze the lifestyles of and promote healthy habits in these students in particular. 

Some studies reported changes in university students’ lifestyles that occurred during home confinement caused by the pandemic [5,6,7,8] but, to our knowledge, no studies have analyzed the lifestyles of these students both before and during the COVID-19 pandemic. Because the return to normal daily living may take some time, understanding the influence that the pandemic exerted on the lifestyles of university students, in particular those enrolled in health sciences programs, may inform strategies to promote their well-being.

## 2. Materials and Methods

### 2.1. Study Design, Participants, and Selection Criteria 

We initiated a longitudinal study between September 2019 and October 2020 that included health sciences students enrolled at the Faculty of Medicine and Nursing, University of the Basque Country, Spain. The selection criteria for participation in the study were: being a nursing or medical student at the beginning of the second year or a physiotherapy student at the beginning of the third year, and to complete the questionnaire in person. In September 2019, we recruited 132 nursing students at the beginning of their second year of the nursing degree program (the response rate was 82%), 128 medical students at the beginning of their second year of the medical degree program (the response rate was 37.5%), and 49 physiotherapy students at the beginning of their third year of the physiotherapy degree program (the response rate was 89.1%). In October 2020, 149 nursing students, 260 medical students, and 47 physiotherapy students completed questionnaires, resulting in 92.5%, 76.0%, and 85.5% response rates, respectively. On all occasions, we gave all students who wanted to participate in the study the opportunity to do so. Thus, the final sample consisted of 113 nursing students, 109 medical students, and 45 physiotherapy students who completed both pre- and post-questionnaires (2019 and 2020).

### 2.2. Data Collection

Data were collected at the Faculty of Medicine and Nursing, University of the Basque Country, Spain. The first data collection period was during September 2019, before the pandemic and implementation of the state of alarm. The second data collection period was in September and October 2020, during the pandemic, when many restrictions were in effect. As a consequence of these restrictions, health sciences students attended combined face-to-face and online classes.

To recruit participants, several information sessions were organized in which students were invited to attend. In these sessions, the objective of the project was presented, doubts were resolved, and their participation was requested. Students interested in participating received the informed consent form and, after their voluntary signature, they proceeded to fill in the questionnaires face-to-face both in September 2019 and October 2020.

### 2.3. Instruments

We used an ad hoc self-administered questionnaire to collect sociodemographic information, such as sex, age, marital status, place of residence during the academic year, number of family members, parents’ educational level, and with whom they lived. 

To assess adherence to the Mediterranean diet (MedDiet), participants completed a 14-item questionnaire validated for the Spanish population and widely used in this type of research around the world, especially in Mediterranean countries [9]. The total questionnaire score ranged from 0 to 14 points, with higher scores indicating higher adherence to the MedDiet. Each item was scored with 0 or 1 point, and 1 point was given for each of the following responses: using olive oil as the main source of culinary fat; consumption of 4 or more tablespoons (1 tablespoon = 13.5 g) of olive oil/day; consumption of 2 or more servings (1 serving = 200 g) of vegetables/day; consumption of 3 or more pieces of fruit/day; consumption of less than 1 serving (1 serving = 100 g) of red meat or hamburger or sausages/day; consumption of less than 1 serving (12 g) of animal fat, such as butter, margarine, or cream/day; consumption of less than 1 glass (100 mL) of sugar-sweetened beverages/day; consumption of 7 or more glasses (100 mL) of red wine/week; consumption of 3 or more servings (1 serving = 150 g) of pulses/week; consumption of 3 or more servings (1 serving = 150 g) of fish/week; consumption of less than two commercial pastries/week; consumption of 3 or more servings (1 serving = 30 g) of tree nuts/week; preferring white meat over red meat; and consumption of “sofrito” (a sauce made with tomato, onion, garlic, or leeks simmered with olive oil) 2 or more times/week. 

Intensity of physical activity was measured using one questionnaire based on the validated International Physical Activity Questionnaire (IPAQ) [10]. We asked participants the following questions: during the last 7 days, on how many days did you do vigorous physical activity (physical activity achieving at least a total of 3000 METs)? During the last 7 days, on how many days did you do moderate physical activity (physical activity achieving at least a total of 600 METs)? To calculate the frequency of physical activity performed we asked participants, how often do you do physical activity in your free time?

### 2.4. Ethical Considerations

This study was conducted in accordance with the Declaration of Helsinki and was approved by the Committee on Ethics in Research of the University of the Basque Country (Humans Committee Code M10/2019/143MR1). 

In accordance with ethical standards for human experimentation, all participants received verbal and written information about the purpose and procedures of the study and signed an informed consent document before beginning.

### 2.5. Statistical Analyses

Statistical analysis was carried out using the IBM SPSS Statistics version 25 statistical software package (SPSS Inc., Chicago, IL, USA). Normality distribution of the quantitative variables was evaluated using the Kolmogorov–Smirnov test. Quantitative variables were presented as mean and standard deviation and qualitative variables were presented using absolute frequencies and percentages. To analyze the difference between MedDiet scores, moderate and intense physical activity, and eating habits before and during the COVID-19 pandemic, a nonparametric bivariate analysis was performed using a Wilcoxon signed-rank test. Paired *t*-tests were used to calculate mean differences (MD) by subtracting before versus during the COVID-19 pandemic with a confidence interval (CI) of 95%, so a negative result reflected an increase in the values. To compare paired proportions between adherence to the MedDiet and physical activity frequency before and after the start of the COVID-19 pandemic, a McNemar test was performed. Statistical significance was determined when *p* < 0.05.

## 3. Results

Participant characteristics are presented in Table 1. The median age of the final sample was 20.19 years. Study subjects included 267 health science university students, of whom 64 were male (24%) and 203 were female (76%). When classifying students according to where they lived and with whom they lived, the majority (70.8%) lived in cities with more than 20,000 inhabitants and with their parents and siblings (74.7%). Most participants were single (86.5%). The vast majority of parents of health sciences students had very high education levels. Specifically, 59.1% of mothers and 45% of fathers had received higher levels of education. 

When assessing participant adherence to the MedDiet, we observed a significant improvement in adherence during the COVID-19 pandemic versus before (*p* < 0.012; MD, −0.240; 95% CI −0.044, −0.436; Table 2). Of note, the mean score before the pandemic was just at the cutoff (9 points) between low and good adherence to the MedDiet. When further analyzing only participants with good adherence to the MedDiet (scores ≥9), we observed that adherence increased from 61.4% to 68.2% when assessed longitudinally (*p* < 0.005).

We found no differences in the frequency or intensity of physical activity (Table 3). Despite slight increases in moderate (*p* < 0.311; MD, −0.050; 95% CI −0.051, −0.151) and intense (*p* < 0.318; MD, −0.043; 95% CI −0.042, −0.128) physical activity during the pandemic, the results are not statistically significant.

Table 4 shows changes in the intake of different food groups. Of note, the consumption of vegetables increased significantly during the pandemic (59.8% to 65.8%; *p* < 0.048). We also observed that the consumption of fatty meats tended to increase (68.8% to 75.9%; *p* < 0.061). We observed the same trend for the consumption of nuts (56.2% to 62.2%; *p* < 0.076). We did not observe significant changes in consumption of the other food groups before and during the pandemic.

## 4. Discussion

We compared university student lifestyles from before the pandemic (September 2019), when all classes were face-to-face, with their lifestyles 1 year later (October 2020), at which time teaching alternated between face-to-face and online classes. One of the most relevant contributions of this study is that it provides a snapshot of the eating habits and physical activity practices at these two points in time, and we observed that during the pandemic, when students had more limited in-person attendance at the university, lifestyle habits, especially in terms of dietary profile, had improved.

The MedDiet, known worldwide for its health benefits, is characterized by a high consumption of fruits and vegetables, olive oil, and nuts. Consistent evidence supports the association of the MedDiet with lower mortality and reduced risk of cardiovascular disease [11,12]. In fact, during the COVID-19 pandemic, the MedDiet was recommended for strengthening the immune system [13]. We found an initial MedDiet adherence score of 9.03 ± 1.69 points among university students, indicating that participants had good dietary habits before the pandemic. Other similar studies examining the dietary habits of university students demonstrated lower or similar scores to those observed in the present study [8,14,15]; however, we did not find any studies demonstrating higher scores. These data are relevant if we compare them with those of the COVIDiet [7] study performed in an adult population during the confinement period in Spain. The scores of our student population were much higher than those observed in the COVIDiet study [15], where participants scored an average of 6.53 ± 2 points before the pandemic and 7.3 ± 1.93 points during home confinement between March and April 2020 [2]. One explanation for these differences between the two studies, as indicated in the DIMERICA study [16], may be that adherence to the MedDiet is affected by different aspects, such as place of residence and education level, and involves not only dietary patterns but also lifestyle, sociocultural, cultural heritage, and environmental aspects. In our study, the percentage of parents with higher education was high, especially for mothers (59.1%). Some articles suggest that the mother’s education level is positively associated with the degree of adherence of their children to the MedDiet [17,18]. In addition, our observations coincide with those reported in other studies suggesting that dietary choices for children and adolescents are made mainly by the mother [18]. Therefore, mothers with higher education levels may influence their children’s dietary choices by making certain foods available and accessible, as well as by being a role model. It is also possible that a higher academic status is associated with higher income and, consequently, a greater availability of healthy foods [19].

When we analyzed the data longitudinally, we observed that both MedDiet adherence scores and the percentage of students with high adherence increased significantly during the pandemic. This increase was mainly due to the increased consumption of vegetables and nuts, which are two of the most characteristic components of the MedDiet. Our results are consistent with most studies [8,20,21,22,23,24] which agree that vegetable consumption increased during the pandemic, although Górnicka et al. (2020) reported a higher percentage of people who decreased vegetable consumption [25]. This increase in vegetable consumption may be related to a higher frequency of food purchase, and it is possible that this increase is also associated with the perception that, in conditions of confinement, it is necessary to eat healthier. In the same sense, other studies [7,8] note that the sale and consumption of legumes, vegetables, and fruits increased significantly after the onset of the pandemic. During the COVID-19 pandemic, people have spent more time at home and take more time cooking than before. In fact, recipe searches on Google skyrocketed [26]. 

As of the time of publication, some restrictions remain in place in Spain, and university students included in this study do not attend university in person every day. We can therefore assume that, by spending more time at home, students or their relatives continue to cook, which is reflected in better eating habits [7]. Of note, one pandemic restriction was the prohibition of dining onsite at the Faculty of Medicine and Nursing of the University of the Basque Country, where this study was performed. In any case, it should be noted that, while the dietary adherence of students included in this study was high before the pandemic, these restrictions on the Faculty of Medicine and Nursing during the pandemic led to significantly improved adherence to the MedDiet. 

We did not observe any variations in physical activity during the pandemic. In other studies, results have been inconsistent, with one reporting an increase in physical activity [16] and another not finding any changes [6]. Therefore, there seems to be no consensus regarding changes in physical activity during the COVID-19 pandemic. 

Several hypotheses could explain these data. One of the hypotheses for observed increases in physical activity is that the environment in which students live affects their sedentary behavior patterns [27]. Rather than being an obstacle, the effect of a restricted social life has benefitted the practice of physical exercise. In the case of health science students, another factor to consider is that their training in promoting healthy habits may have influenced their decision to exercise at home [28]. However, as in our study, the percentage of people who did not engage in physical activity also increased slightly during the pandemic, as did sedentary time [28]. In line with these findings, Di Renzo et al. observed that the lockdown increased activity among people who were active occasionally before the pandemic because they spent more time at home, but those who did not undertake any exercise before the pandemic did not use the situation as an opportunity to start [8].

There is no clear consensus regarding changes in physical activity of students during the COVID-19 pandemic, which reinforces the need for more research. Continued study of the factors related to university students’ physical activity behaviors and their surroundings will help to plan strategies that promote healthy lifestyles.

Our study has some limitations that should be considered. First, dietary and physical activity records were self-reported. Thus, the fact that participants tend to underestimate negative parameters and over-report positive ones must be considered. Second, the conclusions drawn from this sample of university students cannot be applied directly to other populations because not all university students have had the same restrictions as those in the Faculty of Medicine and Nursing. However, this limitation does not preclude reliable results. Third, this study was designed to be conducted face-to-face to ensure the reliability of the reported information. Thus, one of the selection criteria was that the students complete the questionnaire in person, or else unequal attendance in the different grades could have led to a selection bias.

Among the strengths of this study, this is the first longitudinal study that analyzed changes in health science students’ lifestyles as a result of the COVID-19 pandemic. By analyzing the observed changes in diet and physical activity, we can promote activities to improve students’ lifestyles after pandemic restrictions are lifted, and we can devise strategies to adapt in the future, should similar circumstances arise.

## 5. Conclusions

COVID-19 pandemic-related restrictions brought about many changes in the daily lives and habits of health science students in Spain. We estimated the short-term consequences of pandemic restrictions on the lifestyles of university students, but much remains to be investigated. Given the uncertain course of the COVID-19 pandemic in the months and years to come, the findings presented here provide valuable information for planning and implementing interventions aimed at maintaining healthy lifestyles in young adults, and thus preserving their health. Indeed, our results indicate that when students do not have to attend university every day and spend more time at home, their adherence to the MedDiet improves. Strategies should be explored to maintain greater adherence to the MedDiet when the pandemic ends and students return to face-to-face classes. Encouraging and facilitating the consumption of vegetables and nuts at the university could be a good strategy to improve adherence to this diet in the future. To this end, it would be advisable for vending machines in the faculties to always offer the option of nuts and fresh fruit. In addition, university menus should always include vegetable-based dishes at an affordable price for university students. 

Although there were no major variations in physical exercise before and during the pandemic, universities should be concerned about encouraging and facilitating students’ physical exercise habits.

We sought to assess consequences of the COVID-19 pandemic by performing a longitudinal study analyzing changes in the lifestyles of university students enrolled in health sciences programs. Our results provide important information for both public health authorities and educational institutions that should seek to maintain the well-being of students and provide young adults with more opportunities to have healthy lifestyles during their university studies.

## Figures and Tables

**Table 1 nutrients-13-01958-t001:** Participant characteristics.

		Mean (SD)	*n* (%)
Age (years)		20.19 (4.04)	
Sex	Female		203 (76.0)
	Male Total		64 (24.0) 267 (100)
Place of residence	Town < 20,000 population		75 (29.2)
	Town ≥ 20,000 population		182 (70.8)
Marital status	Single		225 (86.5)
	Married		8 (3.1)
	Other		27 (10.4)
Living situation	Alone		5 (1.9)
	Parents and siblings		198 (74.7)
	Other relatives		22 (8.3)
	Other persons		40 (15.1)
Family unit number		3.91 (0.86)	
Access to university	Entrance exam		257 (96.2)
	Other		10 (3.8)
Mother’s education	No studies		2 (0.8)
	Primary		34 (12.9)
	Secondary		72 (27.3)
	Higher education		156 (59.1)
Father’s education	No studies		1 (0.4)
	Primary		41 (15.8)
	Secondary		101 (38.8)
	Higher education		117 (45.0)

Abbreviations: SD, standard deviation.

**Table 2 nutrients-13-01958-t002:** Mediterranean diet (MedDiet) score and adherence (yes/no) before and during the COVID-19 pandemic.

		Before		During		MD	95% CI		*p* Value Wilcoxon Test	*p* Value Student’s *t*-Test
		Mean	SD	Mean	SD					
MedDiet score *n* = 267		9.03	1.69	9.27	1.71	−0.240	−0.436	−0.044	0.012	0.017
		*n*	%	*n*	%				***p* value** **McNemar test**	
MedDiet adherence *n* = 267	**Yes (≥9)**	164	(61.4)	182	(68.2)				0.005	
	**No (<9)**	103	(38.6)	85	(31.8)					

Abbreviations: CI, confidence interval; MD, mean difference; SD, standard deviation.

**Table 3 nutrients-13-01958-t003:** Moderate and intense physical activity (PA) per week and PA frequency before and during the COVID-19 pandemic.

			Before		During		MD	95% CI		*p* Value Wilcoxon Test	*p* Value Student’s *t*-Test
			Mean	SD	Mean	SD					
Moderate PA	*n* = 264		1.00	0.65	1.05	0.65	−0.050	−0.151	0.051	0.311	0.335
Intense PA	*n* = 264		0.98	0.70	1.02	0.70	−0.043	−0.128	0.042	0.318	0.318
			*n*	%	*n*	%				***p* value** **McNemar test**	
PA frequency		**Usually**	137	(51.9)	139	(52.6)				0.182	
	*n* = 264	**Occasional**	116	(43.9)	106	(40.2)					
		**None**	11	(4.2)	19	(7.2)					

Abbreviations: CI, confidence interval; MD, mean difference; SD, standard deviation.

**Table 4 nutrients-13-01958-t004:** Changes in eating habits before and during the COVID-19 pandemic.

		Before		During		*p* Value
		*n*	%	*n*	%	McNemar Test
Olive oil	No	62	23.2%	76	28.5%	0.130
	Yes	205	76.8%	191	71.5%	
Vegetables	No	107	40.2%	91	34.2%	0.048
	Yes	159	59.8%	175	65.8%	
Fruits	No	145	54.5%	148	55.6%	0.813
	Yes	121	45.5%	118	44.4%	
Fatty meats	No	83	31.2%	64	24.1%	0.061
	Yes	183	68.8%	202	75.9%	
Butter. cream	No	35	13.1%	29	10.9%	0.451
	Yes	232	86.9%	238	89.1%	
Carbonated beverages	No	38	14.3%	32	12.1%	0.451
	Yes	227	85.7%	233	87.9%	
Wine	No	241	90.3%	249	93.3%	0.200
	Yes	26	9.7%	18	6.7%	
Legumes	No	135	50.6%	123	46.1%	0.241
	Yes	132	49.4%	144	53.9%	
Fish. seafood	No	153	57.3%	148	55.4%	0.615
	Yes	114	42.7%	119	44.6%	
Bakery	No	84	31.5%	79	29.6%	0.672
	Yes	183	68.5%	188	70.4%	
Nuts	No	117	43.8%	101	37.8%	0.076
	Yes	150	56.2%	166	62.2%	
Lean meat	No	44	16.8%	42	16.0%	0.888
	Yes	218	83.2%	220	84.0%	
Cooked vegetables	No	72	27.0%	67	25.1%	0.653
	Yes	195	73.0%	200	74.9%	

## Data Availability

Data available on request due to restrictions eg privacy or ethical.

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
