# Peer review of "Influence of the COVID-19 Pandemic on the Lifestyles of Health Sciences University Students in Spain: A Longitudinal Study"

_nutrients, 2021, doi:10.3390/nu13061958_

Round 1
Reviewer 1 Report
The study by Imaz-Aramburu et al. is quite innovative and addresses an important topic. Despite being well-presented and structured, I have some concerns that need to be addressed before it can be considered for publication in Nutrients.
Major comments:
- Abstract is general, and does not say much about the outcomes related to influence of the COVID-19 pandemic on the lifestyles.
- I suggest to transfer the information from lines: 64-67 to conclusions.
- The part “1. Study Design, Participants, and Selection Criteria” requires supplementing with selection criteria.
- I suggest to add the information: whether students completed the questionnaire (pre test) stationary face to face or online?
- Was a sample size calculation performed to check the group representativeness?
- The total number of participants and abbreviations (y, SD, n) are missing in the table 1
- The calculations in Table 1 (place of residence, marital status and living situation) and in title of Table 3 do not agree with the total number of participants.
- In order to enrich and improve the discussion (in lines: 210-215) should be divided MedDiet scores (Table 2)according to the socio-economics variables.
- Why the 2 tests (Wilcoxon and t- Student) in Table 3 were used simultaneously?
- Including information (using a separate figure) about the percentage of respondents who met the 1 point criterion for each product groups will improve the readability of the obtained results.
- The conclusions are too general. Specific examples of nutritional interventions should be described (e.g., what the Authors understand writing in line 281: “Encouraging and facilitating the consumption …”?).
Minor comments:
- The title should specify the origin of the students.
- Please avoid repeating words (e.g. lines 35-41, 64-66, 222-223)
- In References there is no literature position by Górnicka et al. (2020), which occurs in lines: 225-229.
Author Response
We enclose the response to Reviewer nº1.
Reviewer 2 Report
Thank you for the opportunity to review this manuscript. The longitudinal design of this study is valuable, as many studies showing dietary changes during the COVID-19 pandemic were performed at only one-time point during the pandemic and only small number of the study investigated longitudinally. However, from the reviewer's point of view, the data were not fully analyzed in the statistical model. In addition, there are some concerns that need to be solved.
Specific points:
- No pandemic may be expected at the time this study was planned (IRB: 10/2019?). Was this study originally planned as a longitudinal study? Was the study design revised due to the COVID-19 pandemic? Please add the detailed study design.
- Lines 69-82: Response rate in the medical student was much lower than that in nursing and physiotherapy students. This may cause the selection bias.
- Tables 2-4: Were there any difference between men and women, or among students major?
- Tables 2-4: Mean difference obtained by subtracting “after value” from “before value” seems to be curious to the reviewer, because “increase” is generally easier to understand when shown as a positive number.
- Table 3: Did the physical activity increase when evaluated as a sum of days of moderate and high levels?
- Table 4: Analyzed variables are nominal scales (0/1). The reviewer does not know using t-tests are adequate.
- Table 4: Many of p-values obtained by Wilcoxon test and T-test were the same. Was it right?
- Table 4: Have all subjects reached the legally permitted alcohol drinking age in Spain? If not so, it is possible that consumption of wine was affected by the participants’ age at the point of the study.
- Lines 258-264: Low response rate in the medical students will be one of the limitation of this study.
- Lines 224-226: In my understanding, there are many studies showing that vegetable intake has decreased during the COVID-19 pandemic.
Author Response
We enclose the response to Reviewer nº2.
Round 2
Reviewer 1 Report
I could not read the Authors' responses to my comments as they are not attached.
However, I see that the changes are applied in the manuscript.
Reviewer 2 Report
No comments.